# Correlation of Soluble CD44 Expression in Saliva and CD44 Protein in Oral Leukoplakia Tissues

**DOI:** 10.3390/cancers13225739

**Published:** 2021-11-16

**Authors:** Ingrīda Čēma, Madara Dzudzilo, Regīna Kleina, Ivanda Franckevica, Šimons Svirskis

**Affiliations:** 1Department of Oral Medicine, Rīga Stradiņš University, 16 Dzirciema Str., LV-1007 Rīga, Latvia; 2Doctoral Study Department, Rīga Stradiņš University, 16 Dzirciema Str., LV-1007 Rīga, Latvia; madara.dzudzilo@rsu.edu.lv; 3Department of Pathology, Rīga Stradiņš University, 16 Dzirciema Str., LV-1007 Rīga, Latvia; Regina.Kleina@rsu.lv; 4Department of Pathology, Children’s Clinical University Hospital, Vienības Gatve 45, LV-1004 Rīga, Latvia; Ivanda.Franckevica@rsu.lv; 5Institute of Microbiology and Virology, Rīga Stradiņš University, 5 Rātsupītes Str., LV-1067 Rīga, Latvia; Simons.Svirskis@rsu.lv

**Keywords:** CD44 antigen, oral leukoplakia, soluble CD44, total protein, exosomes

## Abstract

**Simple Summary:**

One way to facilitate the detection of early malignant transformation processes in oral potentially malignant disorders, including oral leukoplakia, involves the parallel evaluation of tissue and saliva biomarkers, as such indicators may elucidate both protein CD44 expression in leukoplakia tissue and its soluble form and total protein detection in saliva. We concluded that the OncAlert^®^ Oral Cancer Rapid test is a non-invasive and simple but only screening method which can complement clinical diagnostics methods in daily clinical practice when considering oral leukoplakia, most importantly, in order to obtain information about the potential for early malignancy.

**Abstract:**

The aim of this study was to determine whether and how pan-CD44 protein expression in leukoplakia tissues correlates with positive SolCD44 test presence and their role in oral leukoplakia. SolCD44 and total protein expression in saliva were determined using an OncAlert^®^ Oral Cancer Rapid test. Comparison of paired associations of total protein, SolCD44, mean number of CD44 expressed epithelial layers in leukoplakia tissue, and macrophages below the basement membrane between control group and patients with leukoplakia showed statistically significant results (*p* < 0.0001). It is shown that the total protein indicates low or elevated risk of possible malignant transformation processes in leukoplakia. Statistically significant differences between higher total protein level and clinical forms of oral leukoplakia (*p* < 0.0001), as well as CD44-labeled epithelial cell layer decrease (*p* < 0.0001), were found. This possibly points to the onset of the stemness loss in leukoplakia tissue. CD9 antigen expression in the exosomes of the oral epithelium explained the intercellular flow of SolCD44 and other fluids in the leukoplakia area. We conclude that the OncAlert^®^ Oral Cancer Rapid test is a valuable screening method in daily clinical practice, in terms of complementing clinical diagnostics methods and to assess the potential for early malignancy.

## 1. Introduction

In Europe, according to the GLOBOCAN 2020 data, the incidence of lip and oral cancer in both sexes constituted 17.3%, 13.8% mortality, and 5-year prevalence of 20.6% [1]. In the Baltic states, the incidence rates of oral cancer (C03–06) observed in men and women in Lithuania were 2.6 and 0.4; in Latvia, 2.5 and 0.3; and in Estonia, 2.3 and 0.7, respectively [2]. Although the oral cavity is well visible to both the patient and any medical professional, patients tend to seek medical help late when the malignant process already affects adjacent localizations in the mouth. This means that early diagnosis—that is, in the precancerous stage—should be improved. Among oral potentially malignant disorders (OPMD), oral leukoplakia (OL) along with their clinical variants and oral erythroplakia are the most common and have the highest risk of malignant transformation [3].

A systematic review and meta-analysis of the OL incidence in the general population, including the period from 1986 to 2002, revealed that oral leukoplakia was diagnosed in between 1.7% and 2.7% of cases with no geographical differences [4]. Recent research data have shown an increase in OL among OPMD, with a worldwide prevalence of 4.11% [5]. According to data in the literature, 16 to 62% of oral squamous cell carcinomas develops on the basis of oral leukoplakia [6].

Another systematic literature review, including the years 2015 to 2020, assessed the risk factors associated with malignant transformation of OL and estimated the overall malignant transformation of OL presented in different studies as widely ranging between 0.13 and 40.8% [7], with an annual progression rate of 1–3% [7,8]. A systematic review and meta-analysis considering the year 2020 have shown that the malignant transformation rate of leukoplakia is clinically relevant, as 9.5% (99% CI 5.9–14.00%) of lesions developed into cancer with an annual transformation rate of 1.56% [9].

The “gold standard” for determining the risk of malignant transformation involves assessment of epithelial dysplasia [10]; however, there have been intense attempts to find certain markers that could point to the early malignant transformation of oral leukoplakia. In recent years, many studies have been carried out in the search for the most representable tissue molecular biomarkers that could be used to predict the risk of malignant transformation in patients diagnosed with oral leukoplakia [11,12,13,14,15,16].

One of the tissue markers scientists that are still interested in, due to peculiarities in its expression in different cancers (including oral squamous cell carcinoma and its precancerous conditions), is the glycoprotein CD44.

It is known that the CD44 antigen is a multi-structural and multi-functional cell-surface glycoprotein involved in cell-to-cell interaction, cell adhesion, cell proliferation and differentiation, migration, and other biological aspects, and it has also been found in cancer cells [17,18,19].

CD44 is a receptor for hyaluronic acid and can interact with other ligands, such as osteopontin, collagens, and matrix metalloproteinases (MMPs), of which MT1-and MT3- MMPs proteolytically cleave the extracellular stem region of CD44. The cleaved extracellular region is either secreted into the fluid phase or sequestered to the extracellular matrix [20]. CD44 has also been identified as a marker of cancer stem cells (CSCs) in several solid tumors, as well as head and neck cancers, including oral squamous cell carcinomas and epithelial dysplasia [21,22,23,24]. The cancer stem cell model can explain the heterogeneity of cancer cells and facilitate the development of targeted treatment of head and neck squamous cell carcinomas [25].

It has been found that the main ligand of CD44, hyaluronan, is a major constituent of the cancer stem cell niche maintaining the CSC phenotype, which has a substantial impact on the stemness properties of CSCs [26]. The functional significance of different CD44 isoforms has been shown in gastrointestinal, prostate, breast, and pancreatic cancer [26], as well as head and neck squamous cell carcinomas, indicating that the expression of CD44s and CD44v6 is relevant [27]. Nevertheless, there are also controversial studies describing the abundant expression of CD44s and CD44v6 in the great majority of cells in head and neck tissues, including carcinomas; however, CD44s and CD44v6 expression did not distinguish normal epithelia of the head and neck from benign or malignant ones [28]. Other researchers have indicated that among CD44s and its variant isoforms, v5, and v6 may serve as markers for detecting high-risk leukoplakias [29]. It has been suggested that dysplasia and carcinoma are driven by fundamentally different genetic processes [30].

It should be noted that, alongside tissue biomarkers, the study of saliva biomarkers, including the soluble form of CD44, has become a promising diagnostic tool in various general diseases [31], non-oral tumors [32], and oral diseases (especially chronic periodontitis and lichen planus) [22,33,34,35], as well as helpings in early (stage I and II) detection of oral squamous cell carcinoma, oropharyngeal cancers [36], and other head and neck cancers [36,37,38,39,40]. Soluble CD44 can also be detected in serum [41,42] and tears [43]. As mentioned above, there have been several studies on the role of Sol CD44 in the diagnosis of oral squamous cell carcinoma, but fewer on SolCD44 level changes in the saliva of epithelial dysplasia in the case of oral potentially malignant disorders, including leukoplakias. Extensive research on soluble CD44 (SolCD44) in saliva as an early detection tool for head and neck squamous cell carcinoma (HNSCC) has been carried out by Elizabeth J. Franzmann et al. [38,44]. In a 2007 study [44], SolCD44 levels were measured using an ELISA assay, and the authors concluded that SolCD44 in oral rinses when combined with other markers in a panel may serve as a useful marker for the early detection of HNSCC. In 2012 [38], it was shown that the combination of SolCD44 and total protein can improve the sensitivity and specificity of the HNSCC detection test, compared to either marker alone. The results indicated that there is strong evidence that soluble CD44 and total protein are associated with cancer risk independently of tobacco or alcohol use, age, and gender [37]. In the study of Elizabeth J Franzmann and Michael J. Donovan [45], a new method for the early detection of oral and oropharyngeal cancer using a simple and inexpensive Point-of-Care test (OncAlert^®^ Oral Cancer RAPID Test and OncAlert^®^ Oral Cancer LAB Test devices) in oral rinses was discussed. The authors indicated that the OncAlert^®^ Oral Cancer RAPID Test (v2.0) has 90% (79–95%) sensitivity and 62% (53–70%) specificity and recommended it as a Point–of-Care Detection technique in dental or ENT offices complementing the clinical diagnosis.

Saliva is secreted by the parotid, submandibular, sublingual, and minor salivary glands of the oral mucosa. As a result, the saliva is mixed with mucus, inorganic substances, glycoproteins, enzymes, secretary immunoglobulins, lysozymes and non-salivary secretions from gingival crevicular fluid, nasal and bronchial secretions, blood derivates, desquamated epithelial cells, food components, and microorganisms. It also includes proteins, more than 40 kinds, including albumin and globulin [46,47,48].

In recent years, there has been growing interest in exosomes, which have been evaluated as biomarkers for the diagnosis and prognosis of oral diseases, such as squamous cell carcinoma, oral leukoplakia, periodontitis, primary Sjögren’s syndrome, oral lichen planus, and hand, foot, and mouth disease [49]. Exosomes are cell-derived membranous vesicles of endosomal origin secreted by all type of cells and present in various body fluids, such as plasma, serum, human milk, urine, and saliva [50]. Interest in exosomes is due to their biological properties, such as mediating intercellular communication and their capacity to exchange components (e.g., proteins, nucleic acids, and lipids) [51]. A representative which has been used as a marker for exosomes, being contained on their surface, is CD9, a member of tetraspanin family. Tetraspanin proteins are involved in multiple biological processes, including adhesion, motility, membrane fusion, signaling, and protein trafficking, and interact with many different proteins, including interactions between each other [52,53]. Studies have been carried out on CD9 expression in esophageal squamous cell carcinoma and in other cancers [53,54,55]; on salivary extracellular vesicles exosomes in cancers [56]; and systemic diseases [57]. Several studies have analyzed the expression of CD9 in oral squamous cell carcinoma [58,59,60,61]; however, few studies have focused on oral pre-malignant lesions with CD9 antigen expression. We found studies on CD9 expression in epithelial dysplasia [62,63], and a systematic review from the previous year [64] on extracellular vesicles in oral squamous cell carcinoma and oral potentially malignant disorders. There have been studies on the extracellular vesicles in oral saliva of patients with periodontal diseases [65,66], as well as oral squamous cell carcinoma [67].

In this study, we continued to investigate the expression pattern of the tissue biomarker CD44 in homogenous and non-homogenous leukoplakias, as our previous studies have revealed that CD44 intra–cytoplasmatic and membranous expression can be recommended as an early indicator of signs of malignant transformation in non-homogeneous leukoplakia and possible loss of stemness [68]. In this study, we wished to explore whether and how pan-CD44 protein expression in leukoplakia tissues correlates with the positive saliva biomarker SolCD44 test and total protein with its color change on a gradient scale.

## 2. Materials and Methods

### 2.1. Study Group

All patients gave their informed consent for inclusion before they participated in the study. Our study was carried out in compliance with laws and regulations and the ethical principles stated in the Declaration of Helsinki. The study protocol was approved by the Committee of Ethics of Rīga Stradiņš’ University (Decision of the Ethics Committee Nr 3/18.08.2016). The study included 50 patients who were diagnosed with oral leukoplakia in the Department of Oral Medicine, Institute of Stomatology of Rīga Stradiņš’ University and recommended radical excision in the borders of healthy tissues. The operations were performed in the Centre of Maxillo-facial Surgery, of the Pauls Stradiņš’ Clinical University Hospital. The obtained material was subjected to planned morphological studies. Healthy tissue fragments, obtained by excising benign formations in the oral cavity of 20 patients, were used for the control group.

### 2.2. Oral Rinse

Soluble CD44 and total protein in saliva were determined using an OncAlert^®^ Oral Cancer Rapid test (VIGILANT BIOSCIENCES, Ft. Lauderdale, FL, USA). The test device consisted of two strips—one for the CD44 control and test line, and another for the total protein colorimetric assay. A specimen cup and 5 mL of saline were used to collect oral rinse samples. They were collected from patients before biopsy or radical surgical excision. The methodology was followed strictly according to the protocol in all patients. A 5 mL measure of saline was used to collect the oral rinse sample. The patients were asked to rinse for five seconds, gargle for five seconds and then spit into the specimen cup. The test was inserted into the collection cup for three seconds, then laid on a flat surface for 10 min. After 10 min, the results were interpreted. If the control line did not appear, the test was discarded. The test read positive when the CD44 strip showed a line. Total protein results were evaluated based on the gradient scales from 0 to 4. With increasing protein levels, the yellow pad turned from yellow to green, then to blue. The total protein colorimetric assessment was carried out by three clinicians independently.

### 2.3. Microscopic and Immunohistochemical Examination

The resection material of oral leukoplakia and biopsies of the healthy mucosa were completely submitted for microscopic investigation. The tissue samples were fixed in neutral buffered 10% formalin solution. The processed oral mucosa was then embedded in paraplast. Samples from the obtained paraffin blocks were cut into 4 micron-thick sections. The slides were stained with hematoxylin and eosin, then examined under a light microscope. Regarding oral leukoplakia, the following parameters were evaluated: Size and thickness, histological type, degree of dysplasia, and stromal cells under leukoplakia. Presence or lack of small salivary glands in the tissue samples was checked.

Immunohistochemical visualization of the researched antigens was performed on the same formalin-fixed paraffin-embedded oral leukoplakia and control tissue. Immunohistochemical data were collected from the central and peripheral parts of oral leukoplakias.

CD44 and CD9 proteins were assessed by a standard polymer-based visualization En-vision method by Dako Denmark (clone DF1485, dilution 1:50). CD9 antigen was used to detect exosomes. All slides were incubated with 3% H2O2 for 10 min in order to inhibit endogenous peroxidase. Micro-wave-based antigen retrieval was performed in a freshly prepared 0.01 mol/l sodium citrate buffer solution at 750 W for three cycles. Slides were counterstained with Mayer’s hematoxylin.

Cells labeled by CD44 and CD9 antigens displayed staining confined to the cell-surface membrane of oral epithelium and stromal cells under leukoplakia. The number of epithelial layers and mononuclear cells of lamina propria with CD44 and CD9 protein expression was further calculated. Evaluation of both antigens was carried out by two pathologists at the edges and central part of leukoplakias. Immunohistochemical images were taken by the Kappa image base program, using an Axiolab (Zeiss, Oberkochen, Germany) microscope.

### 2.4. Statistical Analysis

Data were tested for the normal distribution using the D’Agostino and Pearson, Anderson–Darling, and Shapiro–Wilk normality tests. Comparisons of the testing groups were carried out by unpaired *t*-tests and one-way ANOVA or repeated-measures two-way ANOVA followed by the two-stage step-up method of Benjamini, Krieger, and Yekutieli as a post hoc procedure. The Spearman’s rank correlation test was used to measure the strength and direction of the associations between the variables. For this analysis, the categorical data were encoded in the following way: Gender (woman, man) assigned as 1 and 2, respectively; levels of the total protein (TP) in four grades (1, 2, 3, 4), assuming 1 and 2 to indicate low, and 3 and 4 to indicate high levels; clinical type of leukoplakia split into three groups (1, 2, 3) assigned as 1—homogeneous form, 2—verrucous and nodular form, and 3—erythroleukoplakia; localization of leukoplakia was characterized by three regions—buccal mucosa and lip (1), tongue and gingiva (2), and floor of the mouth (3). The minimal significance level was set as *p* < 0.05 for all statistical tests. All graphical images and statistical analyses were performed using the GraphPadPrism 9.0 software (GraphPad Software, San Diego, CA, USA) for MacOS. The results are displayed as the median with the interquartile range.

## 3. Results

### 3.1. Clinical, Morphological, and Salivary SolCD44 and Total Protein Characteristics

The mean age of patients in our study group was 57 years (range, 27–82, see Table 1). The patients included *n* = 29 men and *n* = 21, women; thus, the ratio between male and female patients in this study group was 1.3: 1. Oral leukoplakias were mainly localized in the buccal mucosa (*n* = 18), then on the lateral and ventral side of the tongue (*n* = 17), on the floor of the mouth (*n* = 11), and equally on the gingiva (*n* = 2) and lip (*n* = 2). We also noticed that the leukoplakia in our patients was localized more often on the left side of oral cavity (left side *n* = 22; right side *n* = 15), followed by those localized on the floor of the mouth (the anterior part of this localization was more commonly affected).

We diagnosed the following clinical types of leukoplakia: Homogenous leukoplakia in 36% (*n* = 18) and non-homogenous leukoplakia in 64 % (*n* = 32) of patients. Clinically, most were diagnosed as erythroleukoplakia (*n* = 17), then verrucous (*n* = 11) and nodular (*n* = 4) leukoplakia. Analyzing cell and tissue characteristics of surgical material and biopsies, hyperplasia was diagnosed in 17 tissue samples, mild dysplasia in 12, moderate in 8, severe in 10, and intraepithelial cancer in 3 samples.

The OncAlert^®^ Oral Cancer Rapid test revealed that, among patients with homogenous leukoplakia (*n* = 18), the SolCD44-positive test line appeared only in 5 patients, but in non-homogenous (*n* = 32), it appeared in 24 patients (see Table 1).

Total protein results in the case of homogeneous leukoplakia were as follows: color intensity 2 according to the color gradient scale, appeared in 16 patients and color intensity 3 in 2 patients, showing a color change indicating an increase in protein levels. In non-homogenous, leukoplakia-color intensity 2 appeared in 7 patients, while color intensity 3 appeared in 18 patients and color intensity 4 in 7 patients, showing a color change indicating an overall increase in protein levels.

Using Spearman’s rank correlation matrix by analyzing the data of our whole study group (Figure 1), we observed the following tendencies. The number of positive SolCD44 test lines was higher than the value of total protein (TP) (*** *p* = 0.0005) and increased with the severity of the oral leukoplakia clinical form (*** *p* = 0.0006). The number of positive SolCD44 test lines was also increased in cases when the number of CD44-labeled epithelial layers decreased (*p* = 0.0086). Considering total protein (TP), a statistically significant difference was obtained between protein level increase, as evidenced by color change according to the color gradient scale (color 3 or 4), and clinical forms of oral leukoplakia, that is, the higher the protein level, the more severe the clinical form of leukoplakia (*p* < 0.0001). A statistically significant negative correlation was obtained also between higher total protein level and decrease in epithelial layers with positive immunohistochemical expression of CD44 in leukoplakia tissue (*p* < 0.0001). No correlation in the whole study group was found between leukoplakia localization and other studied parameters.

We also observed the tendency that the higher the number of CD44-labeled macrophages under OL, the lower the total protein level, as indicated by lower color gradients on the gradation scale (** *p* = 0.0043).

Studying separately the female group (Figure 2), a statistically significant correlation (**** *p* < 0. 0001) was found between leukoplakia localization and higher total protein level (indicated by color change to 3 or 4).

Analyzing the male group (Figure 3), a statistically significant negative correlation was observed in the terms of total protein level being higher (indicated by color change to 3 or 4), when the amount of CD44 expressed in epithelial layers in leukoplakia decreased (**** *p* < 0.0001).

Figure 4 illustrates the comparison of paired associations between total protein, SolCD44, mean number of CD44 expressed in epithelial layers in leukoplakia tissue, and macrophages in lamina propria between control group and patients with leukoplakia, where *p* < 0.0001 (****) indicates the level of statistical significance of between-group differences, regarding the analyzed parameters.

Comparison of the mean number of CD44 expressed in epithelial layers in leukoplakia tissue and clinical forms of oral leukoplakia showed low correlation (Figure 5).

### 3.2. Immunohistochemical CD44 Antigen Characteristics

CD44 antigen expression was determined in the surgical material and biopsies of oral leukoplakia and healthy mucosa. On average, five basal and intermediate cell layers, without affecting the upper layers of the healthy oral mucosa, showed positive CD44 protein expression. Positive CD44 protein expression was observed only in the cell membrane. CD44 glycoprotein was detected also in the macrophages in lamina propria of oral leukoplakia (Figure 6).

In homogenous oral leukoplakia, the membranous expression of CD44 glycoprotein was in, on average, 19 layers of the epithelium. In non-homogenous oral leukoplakia, CD44 antigen was present not only in cytolemma of the epithelial cells of OL, but also in the cytoplasm of the affected epithelium (Figure 7 and Figure 8). In cases of non-homogenous leukoplakia, CD44 expression was seen in, on average, 15 layers of the epithelium.

### 3.3. Immunohistochemical CD9 Antigen Characteristics

In oral leukoplakias, CD9 expression was detected in the cell membrane at the same areas as CD44 and in an identical number of epithelial cell layers. CD9 was not observed in the upper part of OL and in the places with keratohyalin granules. The pattern of CD44 expression in the cell membrane was like a smooth line, while CD9 antigen presence presented some thickening, causing the membrane to become corrugated (Figure 9).

CD9 protein expression was present also in the ductal epithelium and microparticiples in the lumen of small salivary glands located under the leukoplakia (Figure 10).

## 4. Discussion

Currently, regarding oral leukoplakia, the term “potentially malignant formations” has been suggested, which is more precise as it indicates that not all pathological tissue lesions can turn into cancer [69]. At present, more and more information is being presented on the specific tissues and salivary biomarkers that indicate early signs of malignant transformation of the oral epithelium; therefore, the definition of OPMD will likely need to be reviewed in the future.

It is accepted that oral leukoplakia is two times more likely to be diagnosed in men over the age of 40 (mostly in the 50–70 age group), compared to women, with an average age of occurrence of 60 years, and is six times more common in smokers than non-smokers [70]. Compared to the latest data, in our study, younger patients were already affected, as the patients were 41 to 80 years old, and females were mostly affected [71,72]. The mean age of our patients was 57 years, with the ratio between male and female patients in the patient group being 1.3:1, indicating an increase in incidence in women. We compared our clinical data with literature data based on the classification of leukoplakia from the Western World Patient Data Collection as, in other parts of the world (e.g., in Asia), oral leukoplakia may have different properties, due to different diets, tobacco use, chewing habits, and possibly genetic differences [73].

In the Western population, the most common localization of leukoplakia is on the lateral surface of the tongue and at the base of the mouth, while among the Asian population the cheek mucosa, the vestibular region of the mouth, and the vestibular area in the lower part of the cheek are more affected. In the Japanese population, leukoplakia is more common in the gum area [74]. In our study group, leukoplakias were mainly localized in the buccal mucosa or on the lateral and ventral sides of the tongue, followed by localization on the floor of the mouth and equally on the gingiva and lip. According to other literature data, the tongue (ventro-lateral side) is the most-affected site (55.4%), followed by the buccal mucosa (13.8%), the floor of the mouth (8.8%), gingiva (8.7%), lip (4.1%), and palate (2.2%) [7]. It is likely that, in our patients, the buccal mucosae were affected more commonly due to smoking habits. The tradition of chewing tobacco is not yet widespread in our country, except for foreign students.

The WHO (1978) has recommended distinguishing two clinical types of leukoplakia—homogenous and non-homogenous—with three possible clinical manifestations of the latter—nodular, verrucous, and erythroleukoplakia [73,75,76]. In this study, we diagnosed all of the mentioned clinical forms of leukoplakia.

Among our patients, 88% were smokers. It is well-known that smoking has an effect on the oral leukoplakia microenvironment. Chemicals in tobacco smoke cause chronic inflammation in the oral mucosa, which contributes to an immunosuppressive microenvironment. The results of Yagyuu et al. suggested that non-smoking patients are less likely to develop OL than smoking patients [77]. However, once OL occurs in non-smoking patients, it is associated with a higher risk of malignant transformation. Tobacco smoking is one of the most important risk factors for oral cancer development, regardless of whether it begins as OL or not. The previous studies of Bánóczy et al. also showed that hyperplastic and hyperkeratotic leukoplakia regions have a higher permeability, facilitating the penetration of tobacco carcinogens [78].

In our previous study [68], we noticed that the largest diameter of leukoplakias varied within a relatively wide range (4–30 mm). Of course, if we compare the size of the leukoplakia to the entire area of the oral mucosa, it occupies a small part. According to the studies of Collins and Dawes [79], the mean total surface area of the mouth is 214.7 +/− 12.9 cm^2^, with no significant difference between genders. They indicated that the average volumes of saliva present in the mouth before and after swallowing were estimated as 0.77 and 1.07 mL, respectively. It can be calculated that the average thickness of the salivary film in the mouth varies between 0.07 and 0.10 mm. We are discussing all this as the size and thickness of the leukoplakia may affect the amount of excreted SolCD44 in saliva. In thicker oral leukoplakia, dysplasia will develop with a much higher probability, due to cell organoid injuries, changes in the cell cycle affecting its differentiation and maturation, and impaired tissue microcirculation. The thicker the lesion, the greater the possibility of finding dysplasia in the sample [80]. The areas that showed the greatest permeability (the floor of the mouth and the lateral border of the tongue) corresponded to the regions that are considered “high risk” for squamous cell carcinomas. These areas are more exposed to carcinogens in salivary secretions and the epithelium in this area is more permeable, as indicated by experimental studies considering the oral mucosa. This strongly suggests permeability as a factor in the etiology of malignant oral disease [81].

Martorell-Calatayud et al. indicated that high risk of malignant transformation comprises leukoplakias with mild dysplasia located in high-risk areas, more than 200 mm in thickness, or associated with a non-homogeneous clinical form (i.e., leukoplakias with moderate or severe dysplasia and verrucous leukoplakias) [82].

Some molecular biology studies have recently found that a variable percentage of oral leukoplakias are associated with molecular abnormalities, as has also been found in the case of oral squamous cell carcinoma. These abnormalities reflect an oncogenic potential, regardless of histological atypia. In fact, the appearance of these cytogenetic abnormalities has been described in leukoplakias without cell atypia [82]. It can, therefore, be concluded that the presence of histological dysplasia, regardless of the histological grade, can be considered as an important predictor of malignant transformation of leukoplakia; although histological dysplasia is not necessary for such a transformation to occur.

When analyzing our results, positive CD44 protein expression was observed in the epithelial cell membrane, as well as in the basal and intermediate layers of healthy oral mucosa. As we know, the CD44 protein belongs to the stem cell group, and describes the localization in epithelium, the mediation of adhesive properties, and signals for the orientation of epithelial cells to migrate upward. This coincides with the data of other authors [24,29,83,84,85].

When analyzing leukoplakia tissue samples, the membranous expression of pan-CD44 antigen was seen in homogenous oral leukoplakia; however, in 47% of cases of non-homogenous oral leukoplakia, the expression of CD44 was observed not only in the cell membrane, but also in cytoplasm of the dysplastic epithelium. Most researchers have described the intra-cytoplasmatic pattern of the CD44 expression in real, invasive oral cancers [86], but few have diagnosed its expression in the cytoplasm of the oral epithelium in leukoplakia and explained it considering the interaction of the CD44 antigen with the cytoskeleton [29,87,88,89]. However, other studies are contradictory, for example, the study of Naga et al. [24]. Severe epithelial dysplasia cases showed down-regulated CD44 expression. One of the latest studies by Ghazi et al. [90] analyzed CD44 expression in dysplastic and non-dysplastic oral lichen planus (OLP) tissue. The WHO has classified OLP as a potentially malignant oral lesion, as various data have shown a rather wide range of malignant transformation rates (0–12.5%). The authors found that CD44 was expressed in all lining epithelial cells and sub-epithelial lymphocytes of OLPs, with the greatest total score reported in dysplastic OLP. Additionally, 30% of normal mucosa samples showed negative epithelial immunoreactivity. We observed CD44-labeled macrophages in lamina propria in the control group, but five times more of these labeled cells were found under leukoplakia [68]. Levels of statistical significance, when comparing the mean number of positive CD44 protein expression in epithelial layers in leukoplakia tissue and clinical forms of oral leukoplakia, were weakly expressed (** *p* < 0.01, *** *p* < 0.001).

The CD44 protein has been analyzed in oral mucosa pathologies for over three decades, but recent studies of the CD44 antigen in the oral mucosa have become more relevant, as it can also be detected in saliva. Our results showed that, with an increase in total protein level, the color on the test device changed from distinctly yellow to a gradual intensification of green color, then to green or green with a bluish tinge. These levels were indicated by the numbers 1, 2, 3, and 4, respectively, indicating a low or elevated risk of possible malignant transformation in leukoplakia. A statistically significant difference was found (**** *p* < 0.0001) between higher total protein level (indicated by color gradients 3 or 4) and clinical forms of oral leukoplakia (non-homogenous form, erythroleukoplakia type). The female group revealed a statistically significant (**** *p* < 0.0001) correlation between higher total protein level (color gradients 3 or 4) and leukoplakia localization.

When the SolCD44-positive test lines in our study increased in number, the total protein level increased (color gradients 3 or 4), indicating an elevated risk of malignant transformation and more serious clinical form of oral leukoplakia. When interpreting our data, we followed the recommended methodology of VIGILANTBIOSCIENCES, strictly according to the protocol in all patients. We made sure that one hour before the test, the patient must not have eaten, drunk tea, coffee, or other drinks, smoked, or rinsed his mouth, as this could affect the results.

Our data also showed a tendency of the number of positive SolCD44 test lines to be increased in cases when the CD44 antigen expression in leukoplakia tissue was not only in the oral epithelial cell membrane, but also in its cytoplasm. Similar data were obtained by Erin R. Cohena et al. [91] when immunohistochemically evaluating the CD44 expression in oral and oropharyngeal cancer, as well as SolCD44 in oral rinses; however, it was measured using a sandwich ELISA assay method. The authors identified a significant relationship between SolCD44 levels and CD44 expression in tissue, as well as direct associations between high SolCD44 levels and intense membrane and intra-cytoplasmatic CD44 expression, as we also proved in our study with 50 oral leukoplakias. Significant correlations between the expression of CD44s in biopsy specimens and levels of soluble CD44s in saliva have also been found in OLP patients [34].

When evaluating total protein, it should be taken into account that saliva is a complex fluid containing a wide spectrum of biomolecules, such as proteins/peptides, nucleic acids, electrolytes, hormones from both local and systemic sources, enzymes, antibodies, antimicrobial constituents, and cytokines. Most of the organic compounds in saliva are produced locally in the salivary glands, but some molecules pass into the saliva from the blood. Biomolecules can enter into saliva by diffusion, filtration, and/or active transportation [92]. As previously mentioned, saliva contains secretions not only from (both large and minor) salivary glands and non-salivary secretions, but also desquamated epithelial cells from normal mucosa and leukoplakia tissue. Studies have illustrated the tissue proteome of oral potentially malignant tissues [93,94]. Priya Sivadasan et al. [94] found 73 differentially altered proteins in the saliva of dysplastic leukoplakia. Three proteins—CD44, S100A7, and S100P—were significantly different in dysplastic leukoplakia compared to the normal cohort, with CD44 showing the highest sensitivity. These results indicate that the salivary levels of these proteins can serve as a potential non-invasive screening tool to predict malignancy in oral leukoplakia. Other studies have illustrated the significance of SolCD44 and total protein as useful diagnostic markers for oral, head, and neck squamous cell cancer [37,95].

However, it should be noted that the comparison of paired associations between total protein, SolCD44, and mean number of CD44-positive expression in epithelial layers in leukoplakia tissue between the control group and patients with leukoplakia showed statistical significance (**** *p* < 0.0001). We do not currently have data from other authors who have performed OncAlert^®^ Oral Cancer Rapid testing. However, studies have drawn attention to the fact that SolCD44 detection may help detect dysplasia in tissue (i.e., before clinical evidence). Furthermore, it should be noted that inflammation could lead to false-positive results, as CD44 is a ubiquitous protein [38,44].

Mesenchymal and immune system cells influence the stem cell live factors, but the stemness properties of the mesenchymal stem/stromal cells are limited, and they represent essential components of the stem cell niche and tumor microenvironment [96].

We can assume that, while the stemness of the mononuclear cells—including macrophages—remains and the CD44 marker is expressed on the mononuclear cells, the stemness of the epithelial cells is also controlled. Therefore, the total protein is not high. However, the process by which CD44 proteins lose their stemness properties and, consequently, adult stem cells shift to cancer stem cells, is very fragile and has not yet been clearly explained.

In our study, the tetraspanin CD9 was analyzed, in order to confirm our hypothesis on its participation in two important processes in the oral mucosa. The first one—that this surface protein realizes cell–cell and cell–extracellular matrix interactions—allows for the transportation of Sol CD44 and other fluids to the surface of the oral mucosa. Therefore, it is of scientific and clinical interest that we found that both CD44 and CD9 antigens co-expressed in oral mucosa epithelial cell membranes in both healthy mucosa and oral leukoplakia. The pattern of CD44 and CD9 expression differed: CD44 protein presented as a thin and smooth line along the cell membrane, whereas CD9 led to a corrugated cell membrane, with thickenings that may correspond to exosomes/microvesicles [49,61,92]. CD9-positive microareas of the oral epithelium are possibly so-called transport or channel proteins and explain the process of fluid transport inside the oral leukoplakia.

The second purpose of the parallel examination of CD44 and CD9 proteins was to clarify whether CD9 participates in the transformation processes of oral leukoplakia into cancer. Part of the study groups suggested that a reduction in the quantity of CD9 correlates with progression of oral leukoplakia into a malignant process [62]; however, the study of Wang in 2019 [92] concluded that CD9 inhibits the progression of potentially malignant processes. Our results showed a tendency of decrease in CD9-labeled epithelial layers in non-homogeneous OL, in comparison with homogenous ones, but the result was not statistically reliable. Therefore, the number of biopsies from non-homogeneous leukoplakias for the calculation of CD9- and CD44-positive epithelial layers must be increased in the future. Studies of non-surgically removed leukoplakias and observations of their dynamics are expected to be of great value.

## 5. Conclusions

It should be noted that the OncAlert^®^ Oral Cancer RAPID test is non-invasive and easy to use, which can complement clinical diagnostic methods in daily clinical practice, considering cases of oral leukoplakia, in order to obtain information about the potential for early malignancy. Our study data demonstrated that the total protein level is a possible indicator of early microenvironmental disturbances of leukoplakia, as evidenced by the color change on the gradient scale as the total protein level increased. However, protein evaluation and interpretation should be carried out together with SolCD44 testing. The SolCD44 test line appeared positive when the total protein level and severity of the leukoplakia clinical form increased. CD9-positive microareas of the oral epithelium may serve as so-called transport or channel proteins, as well as explaining the process of fluid transport inside the oral mucosa and leukoplakia. The synchronous quantitative changes of CD44- and CD9-labeled epithelial layers of OL may be related to severe maturation disorders, which are critically important for clinicians and patients in the management of OPMD. From a clinical point of view, the obtained data indicate that the excision of the whole leukoplakia is justified, rather than waiting and observing the dynamics in the patient.

## Figures and Tables

**Figure 1 cancers-13-05739-f001:**
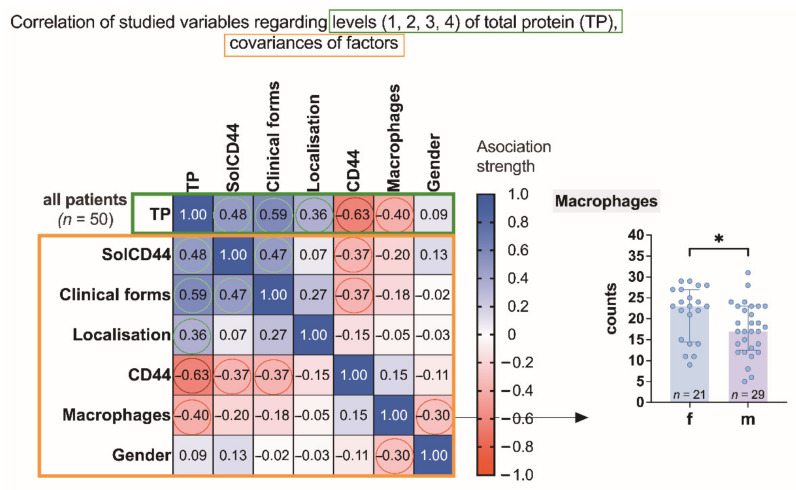
Association strength of studied variables: SolCD44, clinical forms of leukoplakia, localization of oral leukoplakia, mean number of CD44 expressed in epithelial layers in leukoplakia tissue, mononuclear cells (macrophages under the basal membrane), and gender of patients with leukoplakia (*n* = 50) regarding levels of total protein (and covariances). Spearman’s rank correlation matrix: numbers in squared cells show value of correlation coefficient showing association strength; colored circles denote the most significant associations (red, negative; green, positive). Negative association between the number of macrophages and gender (f, female; m, male) is additionally represented by the scatter plot on the right. * *p* < 0.05 (unpaired *t*-test). The arrow indicates a decrease in the average value of the corresponding group.

**Figure 2 cancers-13-05739-f002:**
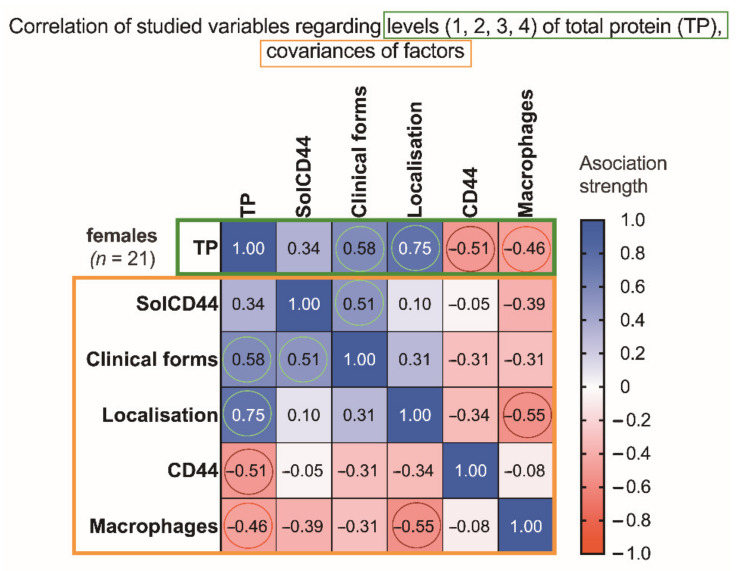
Association strength of studied variables: SolCD44, clinical form of leukoplakia, localization of oral leukoplakia, mean number of CD44 expressed in epithelial layers in leukoplakia tissue, mononuclear cells (macrophages under the basal membrane), and gender of patients with leukoplakia (*n* = 21) regarding levels of total protein (and covariances) in the female group. Spearman’s rank correlation matrix: numbers in squared cells show value of correlation coefficient, indicating association strength; colored circles denote most significant associations (red, negative; green, positive).

**Figure 3 cancers-13-05739-f003:**
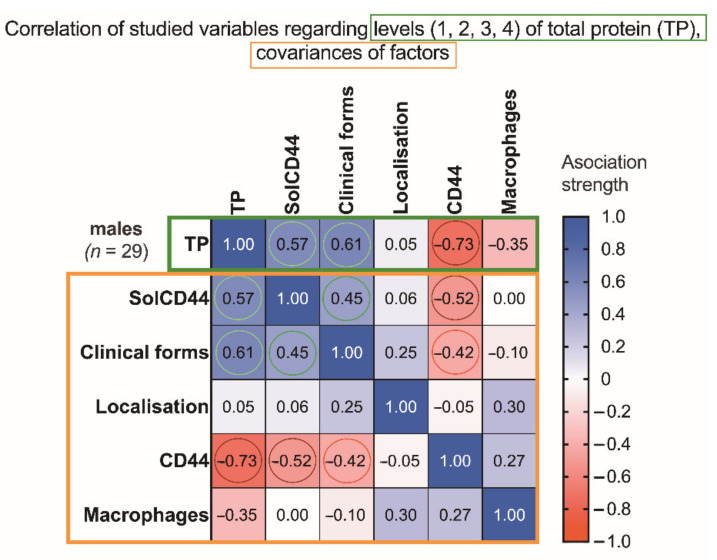
Association strength of studied variables: SolCD44, clinical form of leukoplakia, localization of oral leukoplakia, mean number of CD44 expressed in epithelial layers in leukoplakia tissue, mononuclear cells (macrophages under the basal membrane), and gender of patients with leukoplakia (*n* = 29) regarding levels of total protein (and covariances), in the male group. Spearman’s rank correlation matrix: numbers in squared cells show value of correlation coefficient, indicating association strength; colored circles denote most the significant associations (red, negative; green, positive).

**Figure 4 cancers-13-05739-f004:**
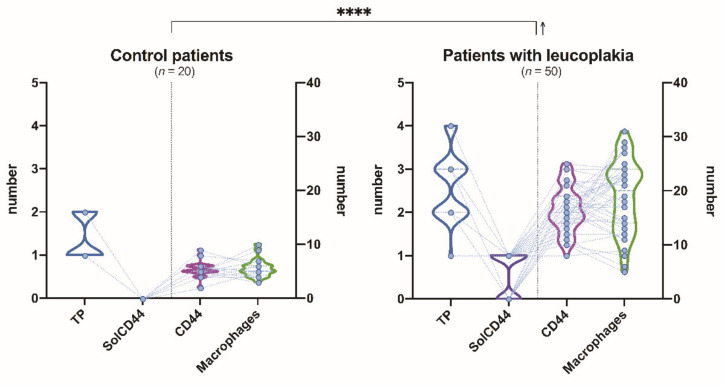
Comparison of paired associations of total protein, SolCD44, mean number of CD44 expressed in epithelial layers in leukoplakia tissue, and macrophages in lamina propria between control group and patients with leukoplakia. **** *p* < 0.0001 (repeated-measures two-way ANOVA with two-stage linear step-up procedure of Benjamini, Krieger, and Yekutieli as post hoc procedure) indicates the level of statistical significance of between-group difference, regarding the analyzed parameters. The arrow indicates an increase in the average values of the corresponding group compared to the control.

**Figure 5 cancers-13-05739-f005:**
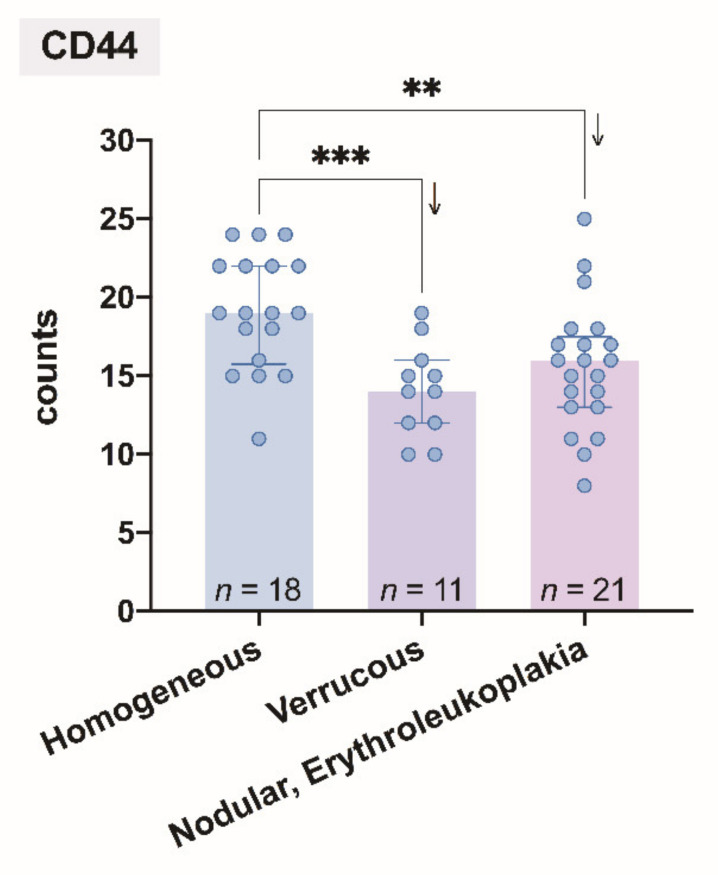
Comparison of the mean number of CD44 expressed in epithelial layers in leukoplakia tissue and clinical forms of oral leukoplakia. Levels of statistical significance: ** *p* < 0.01, *** *p* < 0.001 (Ordinary one-way ANOVA with two-stage linear step-up procedure of Benjamini, Krieger, and Yekutieli as a post hoc procedure). The arrows indicate a decrease in the average value of the corresponding groups.

**Figure 6 cancers-13-05739-f006:**
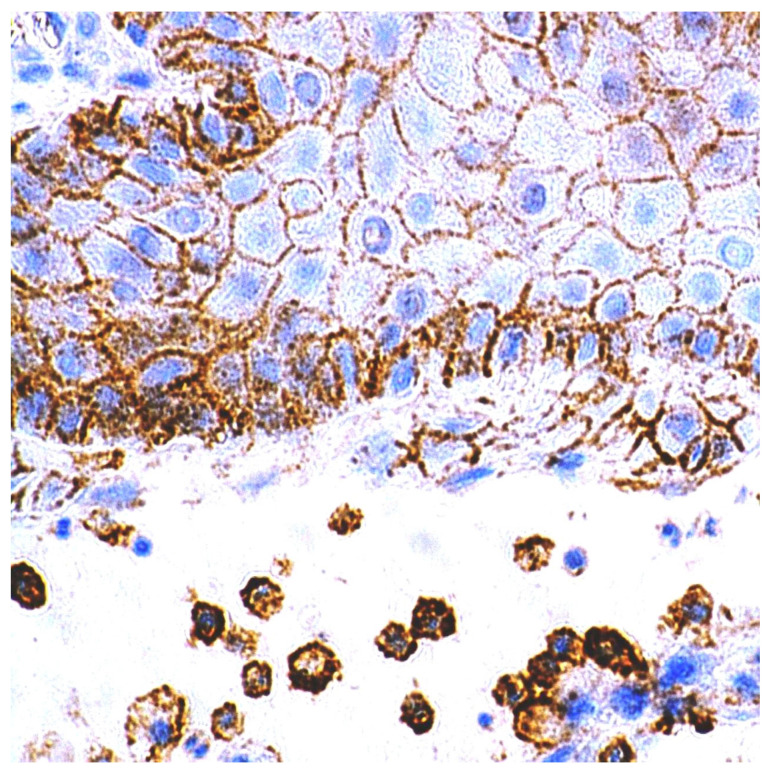
Immunohistochemical visualization of the CD44 marker in epithelium and mononuclear cells of lamina propria of healthy oral mucosa. Immunoperoxidase, anti-CD44, original magnification 400×.

**Figure 7 cancers-13-05739-f007:**
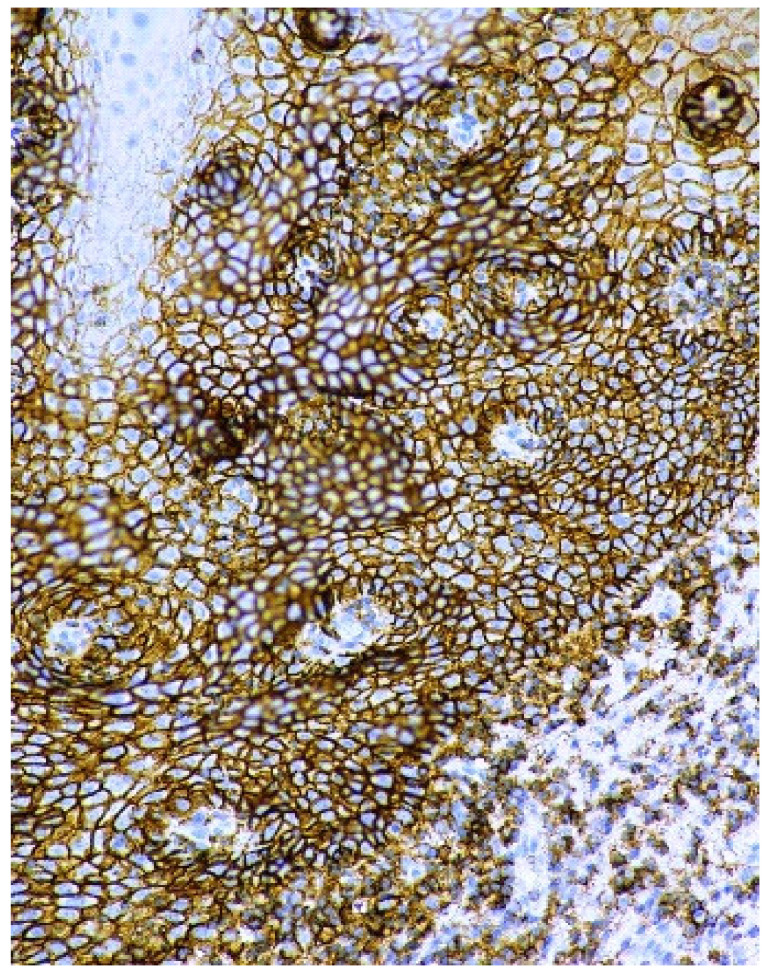
Overexpression of CD44 protein in the cell membranes of epithelium in oral leukoplakia. Immunoperoxidase, anti-CD44, original magnification 200×.

**Figure 8 cancers-13-05739-f008:**
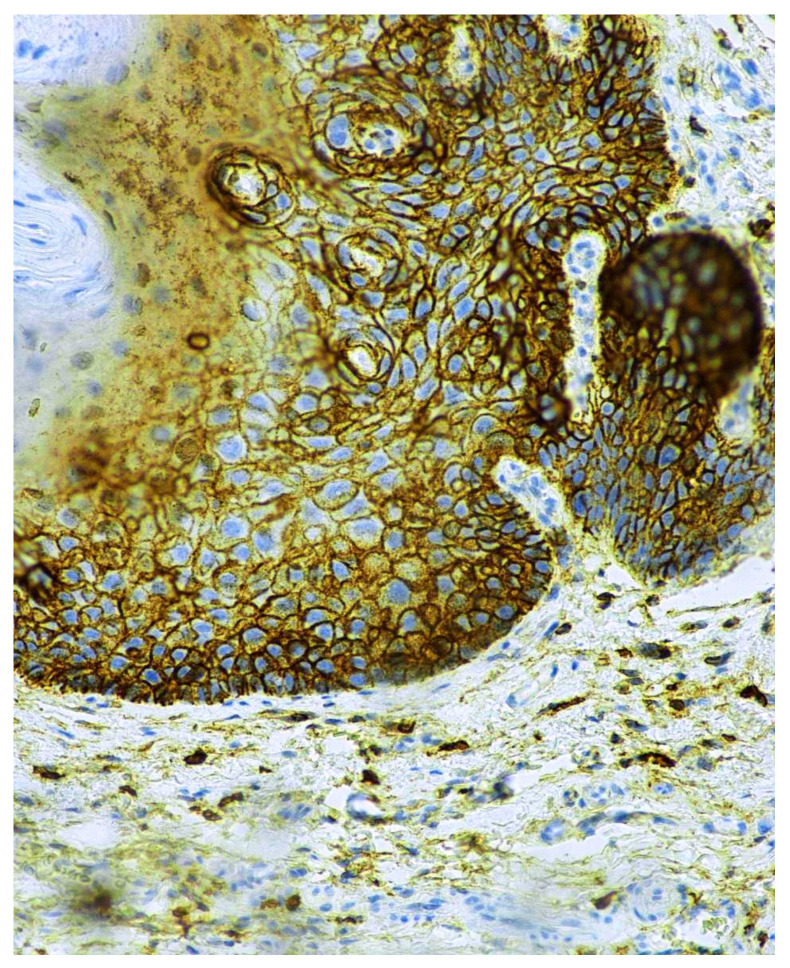
Membranous and intra-cytoplasmatic expression of the CD44 antigen in non-homogenous oral leukoplakia. Immunoperoxidase, anti-CD44, original magnification 200×.

**Figure 9 cancers-13-05739-f009:**
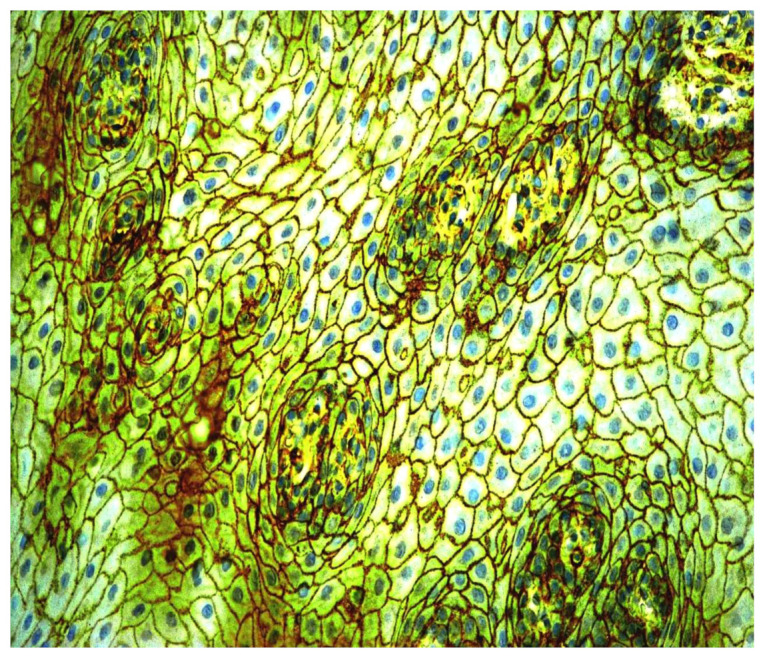
Immunohistochemical visualization of the CD9 marker in the cell membranes of epithelium and its exosomes of oral leukoplakia. Immunoperoxidase, anti- CD9, original magnification 400×.

**Figure 10 cancers-13-05739-f010:**
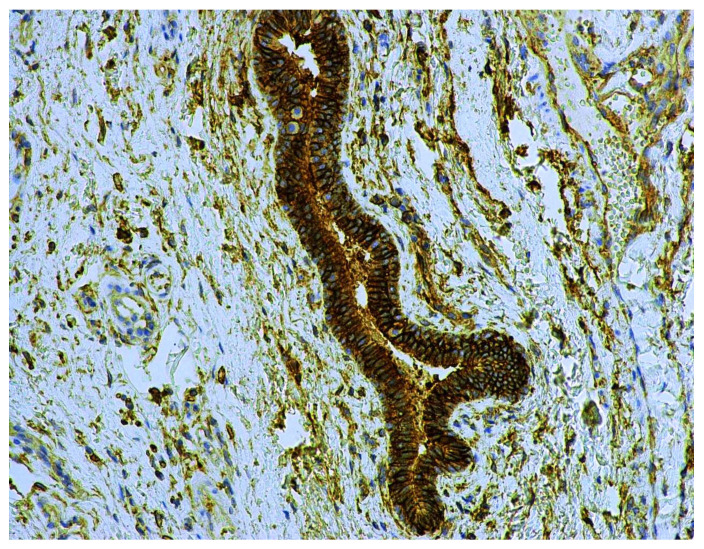
CD9 antigen expression in the lumen microparticiples and the ductal epithelium of small salivary glands under oral leukoplakia. Immunoperoxidase, anti-CD9, original magnification 200×.

**Table 1 cancers-13-05739-t001:** Characteristics of the cohort by age, gender, localization, clinical type of leukoplakia, grade of dysplasia, mean number of CD44-positive epithelial layers in leukoplakia and macrophages in lamina propria, mean number of CD9-positive epithelial layers in leukoplakia, SolCD44, and total protein.

Patient	Age	Gender	Localization	Clinical Type of Leukoplakia	Mean Number of CD44-Positive Epithelial Layers in Leukoplakia (40 × 10)	Mean number of CD9-Positive Epithelial Layers in Leukoplakia (40 × 10)	Macrophages Mean Number, in One Field of View (40 × 10)	SolCD44, Reaction Positive/Negative	Color Intensity of Total Protein (TP)	Grade of Dysplasia in Oral Leukoplakia
1-R	68	F	Fundus cavi oris	Homogeneous	15	16	23	-	high 3	Hyperplasia without dysplasia
2-K	65	F	P. lateralis linguae sin	Verrucous	15	13	29	-	high 3	Hyperplasia without dysplasia
3-P	59	F	P. lateralis linguae sin	Nodular	15	28	14	-	high 3	Mild dysplasia
4-P	68	M	P. lateralis linguae dx	Verrucous	16	17	11	-	high 3	Mild dysplasia
5-E	27	M	Muccosa buccae dx	Erythroleukoplakia	14	15	15	-	high 3	Severe
6-T	56	F	P. lateralis linguae dx	Erythroleukoplakia	15	25	29	-	low 2	Hyperplasia without dysplasia
7-M	50	F	Fundus cavi oris	Erythroleukoplakia	18	19	23	+	high 4	Severe
8-K	62	M	Muccosa buccae sin	Nodular	17	17	17	+	high 3	Mild dysplasia
9-B	40	F	Muccosa buccae dx	Erythroleukoplakia	21	20	23	+	low 2	Mild dysplasia
10-P	59	F	Muccosa buccae sin	Homogeneous	11	16	28	+	low 2	Hyperplasia without dysplasia
11-L	78	F	Fundus cavi oris	Erythroleukoplakia	11	18	21	+	high 4	Severe
12-D	62	F	Muccosa buccae sin	Erythroleukoplakia	16	16	24	+	high 3	Moderate
13-O	55	M	Muccosa buccae sin	Verrucous	15	16	23	+	low 2	Hyperplasia without dysplasia
14-V	65	F	Muccosa buccae dx	Erythroleukoplakia	18	19	22	+	low 2	Hyperplasia without dysplasia
15-P	38	M	Fundus cavi oris	Erythroleukoplakia	13	15	23	+	high 3	Severe
16-S	63	F	Muccosa buccae sin	Nodular	16	16	25	-	low 2	Hyperplasia without dysplasia
17-P	66	M	Fundus cavi oris	Verrucous	19	28	22	+	low 1	Hyperplasia without dysplasia
18-M	37	M	Mucosa proc. Alveolaris mandibulae sin	Homogeneous	19	21	14	+	low 2	Mild dysplasia
19-G	81	M	P. lateralis linguae sin	Erythroleukoplakia	17	17	17	+	high 3	Severe
20-M	56	F	Muccosa buccae dx	Homogeneous	24	27	24	-	low 2	Hyperplasia without dysplasia
21-R	41	M	Fundus cavi oris	Homogeneous	15	17	28	-	low 2	Hyperplasia without dysplasia
22-S	82	M	P. lateralis linguae sin	Verrucous	18	18	12	+	high 3	Mild dysplasia
23-O	79	M	Muccosa buccae sin	Erythroleukoplakia	10	9	8	+	high 4	Ca in situ
24-V	64	M	Muccosa labii inf	Verrucous	10	10	5	+	high 3	Moderate
25-J	29	M	Muccosa buccae dx	Verrucous	14	24	19	+	high 3	Moderate
26-S	34	M	Muccosa buccae dx	Homogeneous	18	20	17	-	low 2	Mild dysplasia
27-F	70	F	Mucosa proc. Alveolaris mandibulae dx	Homogeneous	19	20	27	+	low 2	Hyperplasia without dysplasia
28-K	39	M	P. lateralis linguae sin	Erythroleukoplakia	13	16	23	+	high 3	Severe
29-N	41	M	P. lateralis linguaes sin	Erythroleukoplakia	22	24	17	-	low 2	Severe
30-P	55	M	Muccosa buccae dx	Verrucous	14	18	19	+	high 3	Moderate
31-L	78	M	P. lateralis linguae sin	Erythroleukoplakia	25	25	24	+	high 3	Severe
32-A	63	F	P. lateralis linguae sin	Erythroleukoplakia	17	18	11	+	high 3	Severe
33-S	51	M	Muccosa buccae sin	Homogeneous	22	24	14	-	low 2	Mild dysplasia
34-S	65	F	Fundus cavi oris	Erythroleukoplakia	16	16	11	+	high 4	Severe
35-M	56	M	P. lateralis linguae sin	Verrucous	12	13	12	+	high 3	Moderate
36-S	64	M	Muccosa buccae sin	Homogeneous	16	17	19	+	high 3	Hyperplasia without dysplasia
37-S	29	F	P. lateralis linguae sin	Homogeneous	24	25	22	-	low 2	Hyperplasia without dysplasia
38-M	62	M	Fundus cavi oris	Verrucous	12	10	31	+	high 3	Moderate
39-C	69	F	Fundus cavi oris	Erythroleukoplakia	8	8	9	+	high 4	Ca in situ
40-D	58	F	Muccosa buccae dx	Homogeneous	19	20	27	-	low 2	Hyperplasia without dysplasia
41-D	42	F	Muccosa labii inf	Homogeneous	15	16	28	-	low 2	Hyperplasia without dysplasia
42-K	43	M	P. lateralis linguae sin	Verrucous	10	12	12	+	high 4	Moderate
43-K	49	F	P. lateralis linguae dx	Homogeneous	22	23	14	+	low 2	Mild dysplasia
44-O	63	M	Muccosa buccae sin	Homogeneous	18	19	13	-	low 2	Mild dysplasia
45-D	67	M	Fundus cavi oris	Homogeneous	22	22	18	-	low 2	Mild dysplasia
46-R	66	F	P. lateralis linguae dx	Homogeneous	19	20	15	-	low 2	Mild dysplasia
47-B	49	M	Fundus cavi oris	Erythroleukoplakia	11	10	6	+	high 4	Ca in situ
48-J	77	M	P. lateralis linguae dx	Homogeneous	24	25	24	-	low 2	Hyperplasia without dysplasia
49-M	55	M	P. lateralis linguae sin	Nodular	14	15	23	+	high 3	Moderate
50-L	55	M	Muccosa buccae dx	Homogeneous	22	23	21	-	low 2	Hyperplasia without dysplasia

-: negative; +: positive.

## Data Availability

The data presented in this study are available on request from the corresponding author. The data are not publicly available due to patient data protection law and confidentiality.

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
