# Peer review of "Correlation of Soluble CD44 Expression in Saliva and CD44 Protein in Oral Leukoplakia Tissues"

_cancers, 2021, doi:10.3390/cancers13225739_

Round 1

Reviewer 1 Report

The aim of the study entitled” Correlation of the Soluble CD44 Expression in Saliva and CD44 Protein in Oral Leukoplakia Tissues” was to determine whether and how pan-CD44 protein expression in leukoplakia tissues correlates with positive SolCD44 test presence and their role in oral leukoplakia. The paper show how saliva can be a useful biofluid for the detection of biomarker in oral leucoplakia. The paper can be accepted after minor revision:

Line 138 the salivary protein content shows the presence of much more than 40 proteins, please add the following reference:

Top-down platform for deciphering the human salivary proteome

Castagnola M.,Cabras T.,Iavarone F.,Vincenzoni F.,Vitali A.,Pisano E.,Nemolato S.,Scarano E.,Fiorita A.,Vento G.,Tirone C.,Romagnoli C.

Journal of Maternal-Fetal and Neonatal Medicine. 2012 Volume 25, Issue SUPPL. 5

My biggest doubt relates to the type of assay used to test for total protein (Oral rinse colorimetric assay). Is it a reliable colorimetric test? The detection always remains relative to the observer eye.

Why not use other more effective and accurate quantitative tests for total protein assay?

The total proteins in saliva also follow a circadian rhythm and are also a function of salivary flow. When was the withdrawal made? standardized in all patients? Was the sample processed immediately or stored? Proteins tend to be degraded by salivary proteases. Therefore the correlation with total proteins must therefore be well evaluated and weighted as a parameter

Author Response

Dear reviewer Nr. 1 We have taken into consideration your comments. Thank you.

Point 1: Line 138 the salivary protein content shows the presence of much more than 40 proteins, please add the following reference: Top-down platform for deciphering the human salivary proteome.Castagnola M.,Cabras T.,Iavarone F.,Vincenzoni F.,Vitali A.,Pisano E.,Nemolato S.,Scarano E.,Fiorita A.,Vento G.,Tirone C.,Romagnoli C. Journal of Maternal-Fetal and Neonatal Medicine. 2012 Volume 25, Issue SUPPL. 5

Response 1: We included the recommended article, the reference number 48

Castagnola M, Cabras T, Iavarone F, Vincenzoni F, Vitali A, Pisano E, Nemolato S, Scarano E, Fiorita A, Vento G, Tirone C, Romagnoli C, Cordaro M, Paludetti G, Faa G, Messana I. Top-down platform for deciphering the human salivary proteome. J Matern Fetal Neonatal Med. 2012 Oct;25(Suppl 5):27-43. doi: 10.3109/14767058.2012.714647. PMID: 23025766.

Point 2: My biggest doubt relates to the type of assay used to test for total protein (Oral rinse colorimetric assay). Is it a reliable colorimetric test? The detection always remains relative to the observer eye.

Response 2: In fact, there are no clinical studies on the efficacy of the OncAlert®Oral Cancer Rapid test in cases of oral precancers. Therefore, we want to understand whether this method recommended by VIGILANTBIOSCIENCE can be used as a screening method in the clinic in addition to classical morphological methods, which we have used.  The color change of the total protein in the test was evaluated by 3 authors of this article I.ÄŒ., M.D. and R.K. independently and calculated the average number.

Point 3: Why not use other more effective and accurate quantitative tests for total protein assay?

Response 3: Yes, we agree on the value of using other methods and in the future, we plan to use more effective and accurate quantitative tests for total protein. But in this study, there was a particular interest in the OncAlert®Oral Cancer Rapid test as a clinical screening method in cases of oral leukoplakia.

Point 4: The total proteins in saliva also follow a circadian rhythm and are also a function of salivary flow. When was the withdrawal made? standardized in all patients? Was the sample processed immediately or stored? Proteins tend to be degraded by salivary proteases. Therefore, the correlation with total proteins must therefore be well evaluated and weighted as a parameter

Response 4: We agree that TP may be influenced by many factors but analyzes of its amount was done together with the evaluation of oral leukoplakias morphology and immunohistochemistry. When interpreting our data, we followed the recommended methodology of VIGILANTBIOSCIENCES, strictly according to the protocol in all patients. We made sure that one hour before the test,  the patient must not have eaten, drunk tea, coffee or other drinks, smoked or rinsed his mouth as this could influence the results.We ruled out patients with the diseases of teeth, salivary gland, tonsilla  etc. as an inflammatory processes may affect the amount of total protein in saliva, too.

Reviewer 2 Report

Cema et al. are conducting a  study to improve the preoperative diagnosis of oral leukoplakia at a Latvian university hospital by detecting CD44 and CD9 in saliva.

We believe that this is a very ambitious paper, with appropriate ethical review, patient entry, and analysis, and is an interesting paper that deserves to be accepted.

In reviewing this manuscript, we would like to point out the following items We hope that this paper will be better.

1 The introduction and discussion are often not related to the current study, and I think the whole paper is redundant. I thought the paper would be more readable if it were shaped up to the parts related to the authors' results.

2 The most significant result in Figure 2 seems to be total protein, but it is not clear what TP means in this study. If non-tumor-derived proteins are included in the analysis, I thought that the amount of total protein in saliva may be affected by sampling, such as the patient's oral care status and the time of the last oral intake.

3 The authors seem to have elaborately evaluated CD44 immunostaining for leukoplakia specimens, but did not summarize in figures or tables the relationship between subtype and CD44 and CD9 immunostaining. It would be good to analyze which parameters change most acutely with changes in the type of leukoplakia

4 The macrophage studies are not well discussed in relation to CD44, so it may be an option to exclude them from the results of this article.

Author Response

Dear reviewer Nr. 2 We agree with your comments. Thank you.

Point 1: The introduction and discussion are often not related to the current study, and I think the whole paper is redundant. I thought the paper would be more readable if it were shaped up to the parts related to the authors' results.

Response 1: We agree with your comments about introduction and discussion. We have shortened text specially about oral cancers as it is not the main our article topic.

Point 2: The most significant result in Figure 2 seems to be total protein, but it is not clear what TP means in this study. If non-tumor-derived proteins are included in the analysis, I thought that the amount of total protein in saliva may be affected by sampling, such as the patient's oral care status and the time of the last oral intake.

Response 2:  We agree that TP may be influenced by many factors but detection of it was done together with morphological and immunohistochemical methods examination of leukoplakia. OncAlert®Oral Cancer Rapid test, including Total protein, is only screening method of examination of different oral pathologies. We wanted to check and show how the total protein is overexpressed in patients with oral leukoplakia. When interpreting our data, we followed the recommended methodology of VIGILANTBIOSCIENCES, strictly according to the protocol in all patients. We made sure that one hour before the test,  the patient must not have eaten, drunk tea, coffee or other drinks, smoked or rinsed his mouth as this could affect the results.

Point 3: The authors seem to have elaborately evaluated CD44 immunostaining for leukoplakia specimens but did not summarize in figures or tables the relationship between subtype and CD44 and CD9 immunostaining. It would be good to analyze which parameters change most acutely with changes in the type of leukoplakia

Response 3: In this article we have mentioned that in non-homogenous type of oral leukoplakia “membranous” expression of CD44 was on average, 19 layers of the epithelium but in non-homogenous type - 15 layers. In figure 8 we have demonstrated that CD44 glicoprotein in non-homogenous leukoplakias is detected non only in cell  membrane but  also in cytoplasma of squamous epithelium. But CD9 antigen in this article was used mainly to prove the possible way of transport of saliva constituents through mucosa. We have found that CD9 positive epithelial layers match with the number of CD44 labelled layers and they grow up to 15 layers in non-homogenous leukoplakia.

Point 4: The macrophage studies are not well discussed in relation to CD44, so it may be an option to exclude them from the results of this article.

Response 4: We have removed the description of macrophages from the results and conclusion (lines 302. 303 and line 533. Our research about macrophages in oral leukoplakia cases is published in Proceedings of the Latvian Academy of Sciences. Section B. Natural, Exact, and Applied Sciences.

 (Dzudzilo, Madara, Kleina, RegÄ«na, Čēma, IngrÄ«da, Dabuzinskiene, Anita and Svirskis, Šimons. "Expression and Localisation of CD44 Antigen as a Prognostic Factor of Oral Leukoplakia" Proceedings of the Latvian Academy of Sciences. Section B. Natural, Exact, and Applied Sciences., vol.75, no.2, 2021, pp.68-74. https://doi.org/10.2478/prolas-2021-0012)

This manuscript is a resubmission of an earlier submission. The following is a list of the peer review reports and author responses from that submission.

Round 1

Reviewer 1 Report

The authors used OncAlert®Oral Cancer Rapid test as a non-invasive screening method in cases of oral leukoplakia detecting the soluble form of CD44 and total protein in saliva which are key indicators to assess the potential for early malignancy. In addition, the authors also addressed the negative correlation of soluble CD44 in saliva and CD44 protein in oral leukoplakia tissues for male, but for female. The manuscript although interesting needs further refinement to be considered for publication. The following are comments and questions.

  1. The demographic characteristics of cohort need to be displayed as a table.  
  2. The detected values or information of total protein, SolCD44, CD44, CD9, clinical forms, localisation, gender, and macrophage in each sample need to be provided as a supplementary table.
  3. How to quantify the levels of clinical forms and localisation to get the covariance of variables in figures 2, 3, and 4? The authors need to describe clearly in materials and methods.
  4. In discussion (Line 364~366), the authors mention that the mean age and the ratio between male and female in this cohort showing an increase in the incidence in young people and in women. Please describe detail about how to recruit the subjects into this study? Why the distribution of age and gender in this study reflect the incidence in population?
  5. Will SolCD44 be a more powerful biomarker after normalization by CD44 or total protein in saliva?
  6. The discussion is lengthy and cumbersome, please penetrate into 
    discussion of relative important points.

Author Response

Point 1: The demographic characteristics of cohort need to be displayed as a table.  

Response 1: Demographic characteristics of cohort are shown in one table along with characteristics of the cohort by localisation, clinical type of leucoplakia, grade of dysplasia, mean number of CD44 positive epithelial layers in leukoplakia and macrophages in lamina propria,  mean number of CD9 positive epithelial layers in leukoplakia, SolCD44 and Total protein.

Point 2: The detected values or information of total protein, SolCD44, CD44, CD9, clinical forms, localisation, gender, and macrophage in each sample need to be provided as a supplementary table.

Response 2: We fully accept your remark for the tabular presentation of demographic, clinical, morphological, saliva and biomarker expression rates and macrophage averages of the cohort, but the team of authors decided to present all data in one table. We created a table where characteristics of the cohort by age, gender, localisation, clinical type of leucoplakia, grade of dysplasia, mean number of CD44 positive epithelial layers in leukoplakia and macrophages in lamina propria,  mean number of CD9 positive epithelial layers in leukoplakia, SolCD44 and Total protein are displayed.

Table 1 (please see attachment) shows the working material. A table without coding characters will be inserted in the manuscript.

Point 3: How to quantify the levels of clinical forms and localisation to get the covariance of variables in figures 2, 3, and 4? The authors need to describe clearly in materials and methods.

Response 3: The Spearman rank correlation test was used to measure the strength and direction of the association between the variables. For this analysis the categorical data were encoded in the following way: gender (woman, man) - 1 and 2, respectively; levels of the total protein (TP) - with four grades (1, 2, 3, 4) assuming 1 and 2 to be low, 3 and 4 to be high level; clinical type of leukoplakia was split into three groups (1, 2, 3) and asigned as 1 - for homogeneous form, 2 - for verrucous and nodular one, and 3 – for erythroleukoplakia; localization of leukoplakia was characterized by three regions – buccal mucosa and lip (1), tongue and gingiva (2) and floor of the mouth (3). 

 Point 4: In discussion (Line 364~366), the authors mention that the mean age and the ratio between male and female in this cohort showing an increase in the incidence in young people and in women. Please describe detail about how to recruit the subjects into this study? Why the distribution of age and gender in this study reflect the incidence in population?

Response 4: In this study, we included only those patients who were consulted at the Clinic of Oral Medicine, Institute of Stomatology of Riga Stradins’ University and then surgically treated in the Centre of Maxillo-facial Surgery, of the Pauls Stradins’ Clinical University Hospital.

In April 2020, we planned (at the Clinic of Oral Medicine, Institute of Stomatology of Riga Stradins’ University) an extensive campaign for Latvian residents with the aim of early diagnosis of precancerous conditions. The action was planned for three days, more than 300 inhabitants applied, unfortunately due to the outbreak of Covid -19 infection, this event was canceled. We intend to reorganize it at the end of the pandemic. Since I am for 30 years involved also in the clinical work in Outpatient clinic of RECUH Oncology Center of Latvia, I can certify that late stages of oral cancer are still present. That is why we also planned such a campaign to promote early diagnosis of oral cancer.

Point 5: Will SolCD44 be a more powerful biomarker after normalization by CD44 or total protein in saliva?

Response 5: Our current experience and data suggest that SolCD44 is certainly a good, potent biomarker in daily routine screening practice, both during primary diagnosis and later to follow the course in the dynamics after leukoplakia surgery. Furthermore, SolCD44 reflects epithelial /mesenchimal metabolism of CD44 in oral leukoplakia.

Point 6:  The discussion is lengthy and cumbersome, please penetrate into discussion of relative important points.

Response 6: The discussion we've reviewed, some parts are removed and abbreviated, we consider that the presented one is in a more focused way.

Unfortunately I can't add the edited discussion.

Reviewer 2 Report

The authors recruited 50 leukoplakia patients and 20 normal controls. They detected the Soluble CD44 and total protein in saliva was determined by the OncAlert®Oral Cancer Rapid test (VIGILANTBIOSCIENCES). Immunohistochemistry with CD44 and CD9 antigen were also evaluated in this study.

Comparison of paired associations of total protein, SolCD44, mean number of CD44 expressed epithelial layers in leukoplakia tissue and macrophages below the basement membrane between control group and patients with leukoplakia proved statistical significance (p <0.0001). Statistically significant difference between higher total protein level and clinical forms of oral leukoplakia (p <0.0001), as well as CD44 labelled epithelial cell layer decrease was proved (p <0.0001). The authors claimed that OncAlert®Oral Cancer Rapid test is a valuable in daily clinical practice to complement clinical diagnostics method and to assess the potential for early malignancy.

CD44 in oral cavity cancer were widely investigated in the literature.

The major concern for this study is that all the evaluations of saliva and tissues were done in leukoplakia patients and normal controls. Leukoplakia is a premalignancy but the risk for malignant change is about 1-3%. Not all leukoplakia transforms into cancer. The findings in oral leukoplakia to predict patients’ risk for oral cancer could be over-interpreted. If the authors can recruit a 3rd group of oral cancer group and evaluate the alterations in saliva SolCD44 and tissue expression of CD44. It would be convincing that CD44 is predictive of malignant change.

Author Response

Point 1: The major concern for this study is that all the evaluations of saliva and tissues were done in leukoplakia patients and normal controls. Leukoplakia is a premalignancy but the risk for malignant change is about 1-3%. Not all leukoplakia transforms into cancer. The findings in oral leukoplakia to predict patients’ risk for oral cancer could be over-interpreted. If the authors can recruit a 3rd group of oral cancer group and evaluate the alterations in saliva SolCD44 and tissue expression of CD44. It would be convincing that CD44 is predictive of malignant change.

Response 1: We agree that not all leukoplakias transform into cancer, however in this study we included cases of leukoplakias existing more than two years according to patients disease history data. We wanted to understand what molecular changes are taking place here and we are planning to follow these 50 patients also in dynamics for 5 years. Moreover, our doctoral student M. Dzudzilo, co-author of this manuscript, plans in her theses include a group of patients with oral cancer, so we will be able to  illustrate these results in the next article.

Reviewer 3 Report

The oral leukoplakia is a wide spreading pre-malignant condition of the mouth mucosa. It's important to find and evaluate non-invasive techniques to estimate and monitor the malignant potential of the lesion for the benefit of our patients. This is the main scope of the manuscript offering new information about the use of the OncAlert Oral cancer Rapid test.

The conclusion and the discussion are consistent with the evidence and the references are appropriate.

WELL DOCUMENTED AND VERY INTRESTING MANUSCRIPT. ORAL LEUKOPLAKIA IS AN EVERY DAY CHALLENGE. 

IN THE INTRODUCTION, IS NOT CLEAR THE MEANING OF THE PHRASE:

In Europe it constitutes 17.3 %, with 5 year of survival rate in 20.6%.

Author Response

Point 1: IN THE INTRODUCTION, IS NOT CLEAR THE MEANING OF THE PHRASE: In Europe it constitutes 17.3 %, with 5 year of survival rate in 20.6%.

Response 1: The correct content of the sentence: In Europe, according to GLOBOCAN 2020 data, incidence of  lip and oral cancer both sexes constitute 17.3 %, mortality – 13,8% and 5-year prevalence – 20,6%.  

Round 2

Reviewer 2 Report

The revised article cannot have the 3rd group of cancer patients to validate their observations.